

# Structured peer review: pilot results from 23 Elsevier journals

Mario Malički[1,2,3] and Bahar Mehmani[4]

[1] Stanford Program on Research Rigor and Reproducibility (SPORR), Stanford University, Stanford, CA, United States
[2] Department of Epidemiology and Population Health, Stanford University School of Medicine, Stanford, CA, United States
[3] Meta-Research Innovation Center at Stanford (METRICS), Stanford University, Stanford, CA, United States
[4] STM Journals, Elsevier, Amsterdam, Netherlands

Corresponding author
Mario Malički,
mario.malicki@mefst.hr

## ABSTRACT

**Background:** Reviewers rarely comment on the same aspects of a manuscript, making it difficult to properly assess manuscripts' quality and the quality of the peer review process. The goal of this pilot study was to evaluate structured peer review implementation by: 1) exploring whether and how reviewers answered structured peer review questions, 2) analysing reviewer agreement, 3) comparing that agreement to agreement before implementation of structured peer review, and 4) further enhancing the piloted set of structured peer review questions.

**Methods:** Structured peer review consisting of nine questions was piloted in August 2022 in 220 Elsevier journals. We randomly selected 10% of these journals across all fields and IF quartiles and included manuscripts that received two review reports in the first 2 months of the pilot, leaving us with 107 manuscripts belonging to 23 journals. Eight questions had open-ended fields, while the ninth question (on language editing) had only a yes/no option. The reviews could also leave *Comments-to-Author* and *Comments-to-Editor*. Answers were independently analysed by two raters, using qualitative methods.

**Results:** Almost all the reviewers ($n = 196$, 92%) provided answers to all questions even though these questions were not mandatory in the system. The longest answer (Md 27 words, IQR 11 to 68) was for reporting methods with sufficient details for replicability or reproducibility. The reviewers had the highest (partial) agreement (of 72%) for assessing the flow and structure of the manuscript, and the lowest (of 53%) for assessing whether interpretation of the results was supported by data, and for assessing whether the statistical analyses were appropriate and reported in sufficient detail (52%). Two thirds of the reviewers ($n = 145$, 68%) filled out the *Comments-to-Author* section, of which 105 (49%) resembled traditional peer review reports. These reports contained a Md of 4 (IQR 3 to 5) topics covered by the structured questions. Absolute agreement regarding final recommendations (exact match of recommendation choice) was 41%, which was higher than what those journals had in the period from 2019 to 2021 (31% agreement, $P = 0.0275$).

**Conclusions:** Our preliminary results indicate that reviewers successfully adapted to the new review format, and that they covered more topics than in their traditional reports. Individual question analysis indicated the greatest disagreement regarding the interpretation of the results and the conducting and the reporting of statistical

analyses. While structured peer review did lead to improvement in reviewer final recommendation agreements, this was not a randomized trial, and further studies should be performed to corroborate this. Further research is also needed to determine whether structured peer review leads to greater knowledge transfer or better improvement of manuscripts.

## INTRODUCTION

Journal peer review is a quality control mechanism in scholarly communications that helps editors decide whether to publish a study in the form it was submitted to them, reject it, or recommend revisions or changes before publication. Previous studies, however, have shown multiple deficiencies of the peer review system, (*Tennant & Ross-Hellauer, 2020*) including the failure to detect (significant) methodological deficiencies of articles, (*Schroter et al., 2008*; *Baxt et al., 1998*) spin in results interpretation and generalizability, (*Lazarus et al., 2016*) incorrect use of references, (*Smith & Cumberledge, 2020*) lack of reporting of items needed to reanalyse or replicate studies (*Chauvin et al., 2019*) lack of items needed to assess studies' risk of bias or quality, (*Vesterinen et al., 2011*) *etc.* Studies have also shown (very) low inter-rater agreements between reviewers' recommendations for publication (*i.e.*, 2010 meta-analysis found an inter-rater agreement of 0.34, (*Bornmann, Mutz & Daniel, 2010*) while the most recent Elsevier data covering 7,220,243 manuscripts published from 2019 to 2021 across 2,416 journals, found 30% absolute reviewer agreement for the first review round) (*Petchiappan et al., 2022*).

A part of this disagreement possibly stems from the lack of a pre-specified set of questions reviewers are meant to answer or established criteria used to rate the quality of a manuscript (*i.e.*, an approach that is more common in grant peer review). In August 2022, Elsevier piloted a set of nine structured peer review questions on 220 journals across different disciplines and impact factor quartiles. The questions were implemented in the journal's submission system, but were not mandatory (*i.e.*, each question could be skipped), and except for the last one, which was a drop-down menu, all others were open text fields.

The objectives of this pilot study were to: 1) explore whether and how the first two reviewers of a manuscript utilized those open text answer fields; 2) determine the inter-rater agreements for each question and for the final recommendation; 3) compare the final agreement rates with agreement rates in those journals before implementation of the structured peer review; and 4) further enhance the piloted set of questions. The initial version of this article was published as a preprint at https://www.biorxiv.org/content/10.1101/2024.02.01.578440v1 (*Malički & Mehmani, 2024a*).

## METHODS

Drawing on a previous Elsevier pilot (data not published) on structured peer review in seven journals that contained five questions, MM and BM participated in the creation of a

new question bank (available in full in the Appendix) that editors could adapt and ask reviewers to address during their peer review. In August 2022, Elsevier piloted these nine questions on 220 journals across different disciplines and impact factor quartiles. Before the pilot, the journals did not employ any structured questions. For initial evaluation, we aimed to analyse approximately 10% of that sample of journals and include around 100 manuscripts that received first-round review reports. We therefore obtained data from 25 randomly selected journals across all fields and disciplines. Selecting from that dataset we aimed to include up to initial (first) five manuscripts with two review reports that were received during the pilot period that lasted between 5 August and 5 October. This left us with 23 journals and a total of 107 manuscripts for analysis. The distribution of journals across fields and IF quartiles, and the number of manuscripts per journal, are shown in the Appendix Table 1.

## Review process

Reviewers were asked, but were not mandated, to answer 9 predefined questions regarding: 1) clarity of study objectives, 2) methods reported with sufficient details for replicability or reproducibility, 3) appropriateness of statistical analyses and reporting, 4) changing of tables or figures, 5) appropriate interpretations of results, 6) listing study strengths, 7) listing study limitations, 8) manuscript structure and text flow, and 9) need for language editing. The exact phrasing of the questions is listed in Appendix Box 1. Three of the questions (questions 2, 3, and 5, above) had *Yes*, *No*, or *Not Applicable* (N/A) prompts inserted in an open text field while the rest only had open text fields. Question 9 was unique in this aspect, as reviewers had to choose yes or no using a drop-down menu, with no field to leave additional comments. After these nine questions, the reviewers had the option to leave additional *Comments-to-Author*, and confidential *Comments-to-Editor* by using open text fields. And they could upload documents, *i.e.*, attachments. The text of these two fields indicated that reviewers could add any considerations beyond those listed relevant to the structured set of questions (Appendix Box 1).

Reviewers were also asked to give a final recommendation (*e.g.*, accept, revise, reject, Appendix Table 2). As the database was obtained from the first two months of the pilot and focused on first-round review reports, we did not have access to the editors' final decisions regarding the manuscript. Due to technical aspects, we also did not have access to the authors' uploaded documents, *i.e.*, attachments.

## Answer style coding

We first checked whether reviewers answered questions in their appropriate answer fields, or if they stated that their answer can be found in the *Comments-to-Author* section or in an attachment. As this was simple categorical coding it was done by only one person (MM). We also analysed length (number of words) for each answer (the word count formula we used in MS Excel is presented in Appendix Box 2), as well as for the *Comments-to-Author* and the *Comments-to-Editor* fields. We then checked if those who left *Comments-to-Author* were also more likely to leave *Comments-to-Editor*.

## Comments-to-Author analysis

While reading the *Comments-to-Author sections*, we observed that some reviewers used this field to leave additional comments on top of the answers they left for the nine structured questions (*e.g.*, *In addition, the following aspects must be addressed….*), while others likely copied the full review reports that they prepared beforehand (*e.g.*, such reports often started with a summary of the article or a thank you to the editor for the review invite, and then proceeded with comments on the article). We classified the latter as full (traditional) review reports. We showcase their word count as a separate category, and we also additionally analysed how many of the topics covered by the nine structured questions appear in these reports (*e.g.*, to allow for comparison of traditional reports *vs.* structured reports).

## Answer agreement coding

We independently read and coded answers to each of the questions. The detailed coding book was developed inductively and is available in our data file and Appendix (*Mališki & Mehmani, 2024b*). In short, the main categories we applied were whether reviewer answered the questions with: 1) yes; 2) no; 3) not applicable; 4) yes, and provided comments to the authors on how to address the issue; and 5) no, and provided comments to the authors on how to address the issue. As each question (per manuscript) was answered by two reviewers, we also coded reviewer inter-rater agreement using the following four agreement categories: A) agreement (reviewers had the same answer category as stated above, *e.g.*, 1 to 5); B) disagreement (*e.g.*, reviewers had different answer categories); C) partial agreement (*e.g.*, one review answered yes, while another answered yes, but also provided comments to the authors regarding that question, *e.g.*, category 1 and 4 above); and D) unable to assess agreement (in cases where one of the reviewers did not answer the question, answered N/A, or pointed to the attachment - which we did not have access to). To further elaborate on agreement A, we classified as agreement even cases when both reviewers' answers belong to category 4 above, but their suggestions were not the same (*e.g.*, the first reviewer answered yes to a question, and suggested in the answer field to change the wording to be more reader friendly, while the second reviewer also answered yes, but suggested including a full definition of one of the outcomes). Due to this agreement categorisation choice and for ease of presentation of results we grouped agreement categories A and C when presenting results and labelled them as (partial) agreement in the results, with full results details available in the Appendix.

Initial answers and agreement coding were performed independently by the two coders/authors, after which the coders met regularly (through Zoom) to compare and discuss the codes and agree on the consensus code that would be used in the final analysis. The initial, independent, absolute inter-rater agreement was 94% (Appendix Table 3).

## STATISTICAL ANALYSES

Categories of coded answers and answer styles (see above) are presented as number and percentage calculated based on the number of review reports ($N = 214$), while agreement between reviewers is presented as the percentage of reviewed manuscripts ($N = 107$).

Differences in agreement (for final recommendation and each question) between different scientific fields and journal IF quartiles were assessed with chi-square tests. We also used the chi-square test to compare the final recommendation rates in our study to the agreement rates for the same journals before the pilot (data was provided by Elsevier using the Peer Review Workbench) (*Petchiappan et al., 2022*). The number of words per answer, or text field, is shown as median (Md) and interquartile range (IQR), due to data not being normally distributed. Differences in word count between disciplines and journal IF quartiles were explored with Kruskal–Wallis tests. For comparison with Web of Science/ Publons data (which only showcases the mean number of words for their database), we also showcase the mean number of words in our sample. Additionally, we calculated an odds ratio (OR) and 95% confidence intervals (95% CI) to explore whether reviewers who left *Comments-to-Author* were also more likely to leave *Comments-to-Editor*. All data were explored and analysed using MedCalc v.20.123 (RRID:SCR_015044) or Microsoft Excel v.2211 (RRID:SCR_016137). Significant differences were considered for $P < 0.05$. All statistical outputs from MedCalc were stored in our data repository to help with reproducibility of the results (*Maličcki & Mehmani, 2024b*).

## Study reporting

As the purpose of structured peer review is not only to direct reviewers to focus on issues that help editors make their decisions, but also to serve as a transparent checklist to authors on what their manuscripts will be judged on, we answered the same set of nine questions when preparing this manuscript. Our answers can be found in the Appendix Self-Review section.

## RESULTS

We analysed a total of 214 review reports. These were two first-received review reports for each of 107 original research manuscripts belonging to 23 randomly selected journals across all disciplines and IF quartiles (Appendix Table 1). Each review report consisted of answers to nine structured peer review questions, information left in the *Comments-to-Author* field, and in the confidential *Comments-to-Editor* field.

### Answer style

Almost all reviewers (196, 92%) filled out the answer text fields to all the structured peer review questions (the questions were not mandatory in the system), while 12 (6%) skipped a single question, and an additional 6 (3%) skipped two questions. The most skipped question (by 10, 5% of the reviewers) was question 5 (*Are the interpretation of results and study conclusions supported by the data?*). Some reviewers (16, 7%) directed to attachments or provided an answer and stated that more details were in the attachment (eight did this for a single question, and an additional 8 for more than one question, Appendix Table 4). More than one-third (81, 38%) directed to additional details in the *Comments-to-Author* section or to answers to other questions (most frequently, they did this for a single question (44, 21%). This practice was most common (27, 13%) for question seven (*Have the authors clearly stated the limitations of their study/methods?*). Finally, 38 (28%) left comments in

the answer field of a question that was not related to that question (similarly to above patterns most did this for a single question, 31 (14%), and most frequently, 17 (8%), for question six (*Have the authors clearly emphasized the strengths of their study/theory/ methods/argument?*).

Overall, the combined length of the reviewers' answers to the eight structured questions (the 9th question was a drop-down menu) was 161 words (IQR 73 to 157, with no statistically significant differences between fields, $P = 0.054$, or IF quartiles, $P = 0.268$). The longest answers (Md of 27 words, IQR 11 to 68) were left for question two (*If applicable, is the application/theory/method/study reported in sufficient detail to allow for its replicability and/or reproducibility?*). A detailed analysis of the word count per question, as well as all answer types, is presented in Appendix Table 5.

## Comments-to-Author analysis

Two thirds of the reviewers, 145 (68%), filled out the *Comments-to-Author* section, with no statistical difference between fields ($P = 0.3427$) or IF quartiles ($P = 0.6717$). Of the 145 reviewers that filled out *Comments-to-Author* section, 8 (6%) directed to attachments, 32 (22%) added additional comments or expanded on comments they left while answering the 9 questions, and 105 (72%) provided answers that we classified as traditional full peer review reports (*i.e.*, review reports that were likely prepared beforehand and then copied in the *Comments-to-Author* section*)*. The Md word count of all provided answers in the *Comments-to-Author* section was 323 (IQR 99 to 642), while the Md word count of reports that we classified as traditional full peer review reports was 482 (IQR 282 to 788). For comparison, Web of Science data obtained in June 2023 based on 6,791,816 reviews indicated that the mean word count of all review reports in their database was 368 words (no SD provided). In our sample the mean word count for the full review reports was 580 with SD of 431, but we showed Md and IQR above as data was not normally distributed.

We also checked whether those reports we classified as traditional review reports (see details in methods) addressed the topics covered by the nine structured questions. We found that those traditional review reports ($N = 105$) addressed a median of 4 (IQR 3 to 5) out of 9 questions, most commonly (77, 73%) questions on reproducibility (*e.g.*, details regarding methods), statistical reporting (75, 71%, *e.g.*, additional analyses or analyses reporting) or conclusions and interpretation (68, 65%, *e.g.*, asking for additional discussion points). The east common were requests to list specific limitations (17, 16%) or to emphasize the strengths of the study (1, 1%).

Those that filled out the *Comments-to-Author* section more commonly also left *Comments-to-Editor* (OR = 2.2, 95% CI [1.2–4.2]). In total, 74 (35%) of the reviewers left the latter, with no statistical difference between fields ($P = 0.3976$) or IF quartiles ($P = 0.7339$). The Md word count of the *Comments-to-Editor* was 52 (IQR 23 to 115, with no statistically significant difference between fields, $P = 0.520$, or IF quartiles, $P = 0.448$). Due to the confidentiality of the data, we did not read these comments or analyse them in more detail.

**Table 1 Reviewer (partial) agreement for structured peer review questions of 107 manuscripts.**

| Question | (Partial) Agreement |
|---|---|
| Q1. Are the objectives and the rationale of the study clearly stated? | 58% |
| Q2. If applicable, is the application/theory/method/study reported in sufficient detail to allow for its replicability and/or reproducibility? | 60% |
| Q3. If applicable, are statistical analyses, controls, sampling mechanism, and statistical reporting (*e.g.*, P-values, CIs, effect sizes) appropriate and well described? | 52% |
| Q4. Could the manuscript benefit from additional tables or figures, or from improving or removing (some of the) existing ones? | 64% |
| Q5. If applicable, are the interpretation of results and study conclusions supported by the data? | 53% |
| Q6. Have the authors clearly emphasized the strengths of their study/theory/methods/argument? | 64% |
| Q7. Have the authors clearly stated the limitations of their study/methods? | 67% |
| Q8. Does the manuscript structure, flow or writing need improving (*e.g.*, the addition of subheadings, shortening of text, reorganization of sections, or moving details from one section to another)? | 72% |
| Q9. Could the manuscript benefit from language editing?* | 58% |

**Note:**
*All questions had open text fields, except for question 9 which had a drop-down menu with options of yes or no.

### Inter-rater agreement

Reviewers' agreement per each structured peer review question is presented in Table 1 (with additional details in Appendix Table 6). The highest (partial) agreement (of 72%) was found for assessing the flow and structure of the manuscript, with the lowest (partial) agreement (of 53%) for assessing whether the interpretation of the results was supported by the data, and for assessing whether statistical analyses were appropriate and reported in sufficient detail (52%).

The absolute agreement regarding reviewer final recommendations (exact match of recommendation choice) was 41% (see Appendix Table 2 for all recommendations options in the dataset). No statistically significant difference in absolute agreement was found between scientific fields ($P = 0.5228$) or IF quartiles ($P = 0.2781$). The agreement was greater than that of all journals available in the Elsevier Peer Review Workbench (which had absolute agreement 30%, $P = 0.013$), (9) as well as for only these journals before the implementation of structured peer review (absolute agreement of 31%, $P = 0.0275$ for all manuscripts).

### Question refinement

Based on the pilot analysis, we developed suggestions for improving the questions as they were first posed. For example, while most questions asked if items are clearly stated or appropriately reported/done-where an answer yes would indicate no change is needed, questions four, eight, and nine were opposite, where an answer of yes would indicate that authors need to improve an aspect of the manuscript. Therefore, we modified the questions, so that the answer direction and answer options would be the same across all questions. Examples of additional observations are presented in Appendix Box 3, which also helped guide the refinement of some questions' phrasing. New recommendations for all piloted questions can be found in Appendix Box 4. Finally, we created a new set of eight questions-Box 1 below-with two questions designed for each of the standard sections of a

---

**Box 1 Recommended set of structured peer review questions.**

**Introduction**
1. Is the background and literature section up to date and appropriate for the topic?
2. Are the primary (and secondary) objectives clearly stated at the end of the introduction?
**Methods**
3. Are the study methods (including theory/applicability/modelling) reported in sufficient detail to allow for their replicability or reproducibility?
4. Are statistical analyses, controls, sampling mechanism, and statistical reporting (*e.g.*, *P*-values, CIs, effect sizes) appropriate and well described?
**Results**
5. Is the results presentation, including the number of tables and figures, appropriate to best present the study findings?
6. Are additional sub-analyses or statistical measures needed (*e.g.*, reporting of CIs, effect sizes, sensitivity analyses)?
**Discussion**
7. Is the interpretation of results and study conclusions supported by the data and the study design?
8. Have the authors clearly emphasized the limitations of their study/theory/methods/argument?
**Answer options**
We recommend questions of a multiple-choice style, with the last answer option having a free text field in which reviewers should provide constructive criticism to the authors on how to improve their manuscript. This should allow for easier calculation of rater agreement, and better readability for the authors. We recommend the following choices for all questions:
[ ] Not Applicable
[ ] Beyond my expertise, additional reviewer(s) should be consulted
[ ] Yes
[ ] No, the authors should (consider): (please list and number in the text field below Your suggestions so that the author/s can more easily follow your instructions or provide rebuttals)

---

manuscript (*i.e.*, IMRaD-Introduction, Methods, Results, Discussion). We recommend that editors consider these eight questions for implementation in their journals' peer review practices. If the submission systems allow, we recommend that the questions be implemented as multiple choice questions, where the last option requires text input on the side of the reviewers. Additional questions, specific to other fields or other aspects of manuscripts, can be found on the Elsevier question bank (*Elsevier, 2024*).

## DISCUSSION

Our pilot study of structured peer review implementation has shown that when asked to fill out a set of pre-defined structured questions, reviewers, across all disciplines, responded to those questions in almost equal measure by either adapting their previously prepared peer review reports to answer the questions, or by answering those questions on top of copying their fully prepared reports. Less than 10% of reviewers skipped answering one of the nine structured questions. When we compared the structured questions to the traditional reports, we observed that the reports contained comments for a median of four out of the nine questions asked, indicating that directing reviewers might yield more thorough review reports. Our pilot study, however, was not designed to determine whether the reviewer comments were correct, or to assess their quality, so future studies are needed to determine whether structured peer review leads to easier decision-making choices for the editors, greater improvement of manuscripts, or greater satisfaction of authors in receiving structured reports than in receiving traditional reports.

We have also explored the associations with the reviewer agreement rates. While the final recommendations of the first two reviewers were higher than agreements in those journals during traditional peer review (41% *vs*. 30%), this study was not a randomized

controlled trial, nor was that comparison made with the first two review reports received during traditional peer review (as data were not available). We therefore advise further studies on the influence of structured peer review on the final recommendation agreement, while recognizing that there are differing opinions on whether journals should even ask reviewers for final recommendations or leave that decision solely to the editors. We also observed, that regarding individual questions, the lowest agreement among reviewers was found for what can be considered the core methodological and interpretive aspects of manuscripts, namely whether analyses are appropriate and reported in sufficient detail, as well as are interpretations supported by the data and the analyses. The (partial) agreement rates in our study for these questions were 53% and 52%, respectively. The highest agreement observed was for the manuscript structure, which is likely not surprising, as researchers often follow the structure common in their respective disciplines. As our study was not designed to determine the correctness or reviewers' criticisms and suggestions, or to mediate between their responses and the potential rebuttals from the authors, further studies should explore the factors behind these disagreements, and how the disagreements should be resolved (*e.g.*, in which cases should editors ask from more reviews or side with one of the reviewers). For example, the last structured question used in the pilot was *Could the manuscript benefit from language editing?*, and it had an absolute agreement rate of 58% between the two reviewers (this question had only yes or no as answer options). It is very likely that for this and all other questions/topics, reviewers have different thresholds for the number of mistakes, or readability they find acceptable, and establishing answer option criteria or more uniformed answer categories, might facilitate decision making for editors. We have therefore provided a refined set of questions and answer options and invite editors to consider using or adapting them.

Finally, while this pilot has shown that structured peer review uptake was very high among reviewers, those testing or implementing changes in their peer review processes, should make their reviewers aware in advance of the questions they will be asked to answer and encourage them to answer these questions at the time they are reading and evaluating the manuscripts, and not during the review report submission. Structured peer review may therefore help reduce the "Reviewer 2" fallacy, where the first reviewer provides a very short review and positive review, while Reviewer two provides an extreme opposite. Finally, the authors themselves, if informed on time, might use the same questions as a checklist to self-evaluate and improve their own work.

Our study is not without limitations. This was primarily a qualitative exploration of answer types in an observational study (and not a randomized trial) in which we looked at the direction of the answer, rather than nuances in all answers or number of different suggestions per topic/question that reviewers provided answers to. Additionally, our analysis was based only on full-length research articles, and manuscripts with two reviewers-(dis)agreement rates are expected to be higher for manuscripts with more reviewers. Furthermore, systematic feedback from authors, reviewers, and editors that utilized the structured peer review approach is needed. Additional studies are also needed to determine whether structured peer review leads to greater knowledge transfer or better improvement of manuscripts.

Nevertheless, while we are aware of journals and preprints review platforms that provide a set of structured questions, to the best of our knowledge, this is the first study to explore the answer styles and agreement rates in structured peer review.

In conclusion, this pilot study has shown that structured peer review uptake was very high among reviewers and that agreement rates were higher than in the traditional approach. We would advise those testing or implementing structured peer review to alert reviewers, at the earliest stage possible, about the questions and criteria on how to answer them. Similarly, the authors should be told based on which questions/criteria their articles will be judged and encouraged to use those questions for preparation of manuscripts.

### Funding
There was no additional external funding received for this study.

### Competing Interests
Mario Malički is Co-Editor-in-Chief of Research Integrity and Peer Review journal. Bahar Mehmani is an employee of Elsevier, the publisher of the journals studied in this article and the owner of the submission system used for piloting the structured peer review, and for collecting the data and reviewer responses that were analysed in this study. Elsevier is also the owner of the Scopus database which was used to select journals from different impact factor quartiles and subject areas.

### Author Contributions
- Mario Malički conceived and designed the experiments, performed the experiments, analyzed the data, prepared figures and/or tables, authored or reviewed drafts of the article, and approved the final draft.
- Bahar Mehmani conceived and designed the experiments, performed the experiments, analyzed the data, prepared figures and/or tables, authored or reviewed drafts of the article, and approved the final draft.

### Data Availability
The data and statistical analysis outputs are available at Stanford Digital Repository: Malički, M. and Mehmani, B. (2024). Dataset and Statistical Outputs for Structured Peer Review: Pilot results from 23 Elsevier Journals. Stanford Digital Repository. Available at https://purl.stanford.edu/fn500hr0903. https://doi.org/10.25740/fn500hr0903.

### Supplemental Information
Supplemental information for this article can be found online at http://dx.doi.org/10.7717/peerj.17514#supplemental-information.

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
