# Peer review of "Structured peer review: pilot results from 23 Elsevier journals"

_PeerJ, doi:10.7717/peerj.17514_

## Round 0.1 · original submission · Minor Revisions

Thanks for your comments. The reviewers found much to like in your paper, while also providing some constructive feedback. Please address all of their comments, along with these additional points:

1. Did the analysis of the free-text answers suggest any possible revisions or extensions to the structured review?

2. Under Methods, you mention "to 5 first manuscripts". What is meant by a "first manuscript"?

3. Given that questions 2,3, and 5 were also yes/no/Non-applicable, why weren't those questions implemented as drop-downs, like question 9?

**Language Note:** The review process has identified that the English language must be improved. PeerJ can provide language editing services - please contact us at [email protected] for pricing (be sure to provide your manuscript number and title). Alternatively, you should make your own arrangements to improve the language quality and provide details in your response letter. – PeerJ Staff

·

Basic reporting

I recommend that the manuscript be read by a fluent English speaker to check the need for indefinite and definite articles, as well as the structure of many sentences where there is heavy reliance on describing information in parentheses. For example, the authors state on lines 264-266, “For comparison, Web of Science data indicates that the mean word count of all review reports in their database is 368 Words (June 2023 based on 6,791,816 reviews, no SD is provided).” The sentence should also be in past tense. This sentence is awkward and lacks flow and clarity and could be re-stated as “For comparison, Web of Science data from June 2023 based on 6,791,816 reviews indicated that the mean word count of all review reports in their database was 368 words (no SD provided).

Experimental design

The authors' pilot study regarding questions for structured peer review is a great start to help editors provide guidance to their pool of reviewers. Structured peer review would be great to facilitate reviewers' responses, where future studies could determine the appropriateness and quality of those reviews.

Validity of the findings

Methods
Statistical analyses
Line 215 – The Kruskal-Wallis test was performed, but could the authors mention which post-hoc test (Conover’s, perhaps) was used to assess specifically between which disciplines and IF quartiles were the word counts different? I did not see any post-hoc tests in the statistical test outputs in the file entitled, “Statistical Outputs for Structured Peer Review Pilot results from 23 Elsevier Journals.docx” available in the Stanford Digital Repository. MedCalc automatically provides Conover’s post-hoc tests, but these are not provided in this document. Could you please re-submit a revised output document where the post-hoc test results are provided?
Lines 217-218 –Please state the abbreviation for the odds ratio (OR) and that 95% confidence intervals (CI) were described for them.

Additional comments

Minor
Line 151 – Please change “Non Applicable” to “Not Applicable.”
Line 192 – Please change “NA” to N/A.”
Lines 194 and 353– Please change “reviewers” to “reviewers’” to express possession rather than the plural form of the word.
Line 195 – Please add indefinite and definite articles where appropriate throughout the manuscript. Several examples where indefinite or definite articles are missing include lines: 197 (needs “a” before “full”), 208 (needs “the” before “percentage”), 238 (needs “an” before additional), 241 (needs “an” before “additional”)… there are many instances throughout the manuscript.
Line 195 – “Are” should be past tense as “were.”
Line 213 – Define “Md” and “IQR” at their use as “Median (Md)” and “interquartile range (IQR)”
Line 276 – Please spell out “stat.” throughout the manuscript to be “statistical” where applicable.
Line 332 – “Satisfactions” should be singular as “Satisfaction”.

·

Basic reporting

This research article represents a pilot study to check how some changes/tools in the peer-review process affect the reviewers` actions.
Although valuable results were obtained it seems that one of the main comparisons has been omitted in the study. That is the comparison of the final suggestions/recommendations of the reviewers after the implementation of the structured review versus their recommendations before that. This variable can provide data on whether the structured reports affect the decisions of the reviewers in terms of their final suggestions to the editor.
It can be also concluded from the content (defined as a traditional report) that before the implementation of this pilot study the editor did not require any kind of form from the reviewer but the reviewers were free to express their comments in the format they considered suitable. If this is the case, it should be clarified in the study not left to the readers to assume this.

Experimental design

no comment

Validity of the findings

no comment

Additional comments

no comment

---

## Round 0.2 · Minor Revisions

Thanks for the revisions. You have addressed the comments from the reviewers, along with most of my comments. There is one outstanding clarification that you addressed in the response, but not in the manuscript. In the first paragraph of the methods, your mention of the "5 first manuscripts" is still unclear. Please revise this sentence with language similar to what you used in the response letter.

---

## Round 0.3 · accepted · Accept

Thank you for addressing the outstanding concerns.